# Porcine Relaxin but Not Serelaxin Shows Residual Bioactivity after In Vitro Simulated Intestinal Digestion—Clues for the Development of New Relaxin Peptide Agonists Suitable for Oral Delivery

**DOI:** 10.3390/ijms24010048

**Published:** 2022-12-20

**Authors:** Lorenzo Pacini, Annunziata D’Ercole, Anna Maria Papini, Daniele Bani, Silvia Nistri, Paolo Rovero

**Affiliations:** 1Interdepartmental Research Unit of Peptide and Protein Chemistry and Biology, University of Florence, 50019 Florence, Italy; 2Department of Chemistry ‘Ugo Schiff’, University of Florence, 50019 Florence, Italy; 3Research Unit of Histology & Embryology, Department of Experimental & Clinical Medicine, University of Florence, 50139 Florence, Italy; 4Department of NeuroFarBa, University of Florence, 50139 Florence, Italy

**Keywords:** serelaxin, porcine relaxin, proteolysis, intestinal digestion, RXFP1, THP-1 cells, cAMP

## Abstract

Despite human recombinant H2 relaxin or serelaxin holding promise as a cardiovascular drug, its actual efficacy in chronic treatment of heart failure patients was hampered by the need to be administered by multiple daily IV injections for a long time, with obvious drawbacks in terms of patients’ compliance. This in vitro study aimed at exploring the molecular background for a possible administration of the peptide hormone relaxin by the oral route. Serelaxin and purified porcine relaxin (pRLX) were subjected to simulated intestinal fluid (SIF) enzymatic digestion in vitro to mimic the behavior of gastroprotective formulations. The digestion time course was studied by HPLC, and the relative bio-potency of the intact molecules and their proteolytic fragments was assessed by second messenger (cAMP) response in RXFP1 relaxin receptor-bearing THP-1 human monocytic cells. Both intact proteins (100 ng/mL) induced a significant cAMP rise in THP-1 cells. Conversely, SIF-treated serelaxin showed a brisk (30 s) bioactivity decay, dropping down to the levels of the unstimulated controls at 120 s, whereas SIF-treated pRLX retained significant bioactivity for up to 120 s. After that, it progressively declined to the levels of the unstimulated controls. HPLC analysis indicates that this bioactivity could be ascribed to a minor component of the pRLX sample more resistant to proteolysis. When identified and better characterized, this peptide could be exploited for the development of synthetic relaxin agonists suitable for oral formulations.

## 1. Introduction

Despite favorable premises collected from many preclinical studies, the cumulative results of Phase III clinical trials with serelaxin (RELAX-AHF-2, ClinicalTrials.gov No. NCT01870778), the recombinant form of human H2 relaxin (RLX), a luteal hormone with well-assessed cardiovascular and anti-fibrotic effects [1,2,3,4], for the treatment of acute heart failure patients failed to meet the expectations. In fact, serelaxin, given by IV infusion (30 μg/kg/day), although it improved some renal function markers, showed no substantial effects on length of hospitalization, occurrence of cardiovascular death, and disease relapses requiring re-hospitalization in comparison with the placebo-treated group [5,6]. In a reappraisal of these studies, some key limitations of the adopted therapeutic protocol have been pointed out as a possible cause for such failure. The most relevant is that serelaxin has a short half-life, about 2 h [7], and its in vivo effects rapidly vanish [8]. Hence, to achieve steady therapeutic circulating levels, serelaxin has to be administered IV in repeated daily doses, with obvious issues in terms of patients’ compliance. This led the experimenters to limit serelaxin infusion over the first 48 h from admission to the hospital, while patients’ data were collected far later, namely at 60- and 180-day follow-up, albeit its beneficial effects appeared to be lost by day 14 post-therapy [5,6].

This recent paradigm highlights the need to explore alternative modes and routes of administration for serelaxin, in order to allow this peptide drug to achieve therapeutic levels and effects even in diseases, that require enduring treatment, for instance, upon hospital dismissal or for primary and secondary prevention in subjects at risk. A first-line, logical choice could be implantable infusor devices, like those currently under study for insulin delivery in type-1 diabetic patients [9], although they are expensive and not free from side effects [10]. Another possibility could be to develop pharmaceutical formulations of serelaxin suitable for oral delivery: Such kind of medication would definitely increase the patients’ liking and willingness to commence and maintain the therapy. The major challenge of this approach is that the serelaxin peptide must be made able to withstand gastrointestinal digestion and cross the gut-blood barrier in pharmacologically effective amounts [11]. Of note, such kind of issues have been addressed by researchers in the pharmaceutical field for the set-up of oral insulin preparations to increase the therapeutic options for diabetic patients [12,13]. This scientific know-how could be easily transferred to serelaxin, owing to the substantial molecular similarities between the two hormones [14].

Another variable comes from the possibility that proteolytic serelaxin fragments generated by gut enzymes may retain the capability to bind to and activate the RXFP1 RLX receptors on target cells upon having reached the bloodstream. This is substantiated by the observation that H2 RLX derivatives with truncations at A and B chain termini retain RLX-like activity and can be actually considered RLX agonists [15].

The present study was designed to provide additional insight into the existing knowledge on the feasibility of oral administration of 2 RLX molecules used previously for therapeutic purposes, namely serelaxin and purified porcine RLX (pRLX), using an in vitro model to simulate the possible behavior of gastroprotective formulations subjected to gut enzymatic digestion. The time course of the digestion was studied by High-Pressure Liquid Chromatography (HPLC), and the relative bio-potency of the intact molecules in comparison with their proteolytic fragments was assessed by measuring the signaling events downstream receptor activation in THP-1 human monocytic cells. These cells were chosen as a suitable model to assess the biological response to serelaxin because they constitutively express RXFP1 receptors, whose stimulation results in prompt activation of the adenylate cyclase signal transduction pathway and cAMP rise [16,17].

## 2. Results

### 2.1. cAMP Response by THP-1 Cells

Stimulation of THP-1 cells for 15 min with intact serelaxin (100 ng/mL) or pRLX (100 ng/mL) induced a statistically significant increase in intracellular cAMP in comparison with the controls. Such effect attained, respectively, 88.6% and 68.4% of the maximal cAMP surge induced by the adenylate cyclase activator forskolin (10^−4^ M). The higher efficacy of serelaxin with respect to pRLX well correlates with its higher affinity for the RXFP-1 receptor demonstrated in similar cellular models [18]. Instead, the digestion products of serelaxin showed a progressive decrease in cAMP production which correlated with the time of exposure to SIF. This effect was already appreciable at 10, 20, and 30 s. and became fully manifest and statistically significant at 60 and 120 s: at this latter time point, the cAMP levels dropped down to similar levels as those detected in the unstimulated control cells. Conversely, the digestion products of pRLX did retain the capability to stimulate cAMP generation by THP-1 cells at levels comparable to those achieved by the intact peptide at any SIF exposure time tested, e.g., up to 2 min (Figure 1).

On this ground, we further explored the bioactivity of pRLX digestion products at longer times, namely up to 30 min, to identify their maximal lifespan. These additional results showed that a marked decay of cAMP levels by THP-1 cells occurred between 2 and 5 min, reaching values not significantly different from the unstimulated controls (Figure 2). In parallel control experiments, we also ascertained that SIF alone had no cAMP-stimulating effect on the same cell model (Figure 2).

### 2.2. Time Course of pRLX Digestion

When setting up the chromatographic method to follow the time course of pRLX digestion, we observed heterogeneity of the starting product. In fact, the sample of pRLX, dissolved in water and analyzed at time 0, i.e., before adding SIF, produced the chromatogram shown in Figure 3, panel 1, clearly displaying the main peak (Rt: 7.700 min) and three minor ones (Rt: 7.596, 7.917, and 8.025 min), labeled A, B, C, D in Figure 3. This observation is in line with early literature reports, indicating that pRLX extracted from sow ovaries is heterogeneous [19,20]. In particular, Kohsaka and co-workers [20] reported the presence of four isoforms of relaxin in ovaries of pregnant sows, endowed with similar chromatographic and electrophoretic behavior and, most importantly, similar biological activity. Interestingly, the HPLC elution profiles of these isolated isoforms closely resemble those reported herein (Figure 3, panel 1). The time course of pRLX digestion indicates that the main peak B disappeared after 10 sec of enzymatic SIF digestion (Figure 3, panel 2), while the other peaks appeared to resist longer to the enzyme digestion. In particular, peak A is present up to 2 min (Figure 3, panel 3), and peak C, with a shoulder corresponding to peak D, is still present at t = 5 min (Figure 3, panel 4) and even at longer t = 10 min. The close correspondence between the timing of the disappearance of peak A in the chromatogram and of the cAMP-stimulating effect strongly suggests that peak A should coincide with the SIF-resistant fragment with RXFP1 agonist properties.

## 3. Discussion

Pharmaceutical research is facing an increasing demand for effective formulations and methods for oral administration of protein therapeutics, which represent a significant part of the new drugs approved for the market in recent years. These include hormones, such as insulin, glucagon, calcitonin, GH, and gonadotropin-releasing factors, and biologicals, such as interferons and other immune modulators, monoclonal antibodies, enkephalins, vaccines, enzymes, and enzyme inhibitors [21]. Protein therapeutics are particularly attractive because they behave as highly selective, oligodynamic drugs, often endowed with fewer side effects than chemicals and capable of targeting the causative mechanisms of diseases rather than alleviating their symptoms [22].

The major challenges in the development of efficacious oral peptidic drugs, which would be desirable for multiple daily and/or long-term treatments, depend on their chemical nature, which renders them prone to rapid catabolism and impedes effective absorption and biodistribution. Protection from salivary and gastric digestion can be easily achieved by appropriate oral/gastro-resistant formulations designed to dissolve in the gut, whereas intestinal digestion cannot be prevented completely as the active peptide has to be released and absorbed in the gut [21,22].

The current study was specifically designed to investigate this issue, namely whether two similar RLX molecules suitable as protein therapeutics, serelaxin and pRLX, could be subjected to simulated intestinal digestion while retaining yet detectable biological effects. The reported findings showed that both the intact proteins at the known bioactive concentration of 100 ng/mL induced a marked, significant rise of cAMP in RXFP-1-expressing THP-1 cells. Instead, exposure of the serelaxin solution to SIF caused a substantial loss of bioactivity, in terms of detectable intracellular cAMP, already appreciable after the shortest 30 s incubation and dropping down to the levels of the unstimulated control cells after 120 s. Conversely, and unexpectedly, the SIF-exposed pRLX samples retained full capability to stimulate cAMP generation by THP-1 cells for up to 2-min digestion, after that, it rapidly declined to levels similar to the untreated controls. HPLC analysis indicates that this stimulatory activity should be ascribed to the minor component of the pRLX sample, corresponding to peak A in our chromatogram. Additional studies are undoubtedly required to identify the amino-acidic sequence of this molecule, as such knowledge could be exploited for the development of synthetic relaxin agonists suitable for oral formulations. In this context, we have recently shown that human H1 RLX analogs can be stabilized by intra-chain tri-azolic staples in the appropriate 3-D conformation in order to further enhance their resistance to digestive enzymes [23]. Our forthcoming study will explore whether such chemical modification can also be applied to the pRLX component to obtain an effective and digestion-resistant RLX agonist.

Taking into account the limitations of the present experimental setting, which is a simplification of the complex digestion and absorption events occurring in vivo, the reported findings provide serious clues against the possibility of developing effective serelaxin preparations to be administered orally, since susceptibility of H2 RLX to be rapidly decomposed by gut digestive enzymes is particularly high, conceivably due to species-specific homology between the peptidic substrate and the proteolytic enzyme, and the putative peptidic remnants appear to be devoid of any residual tropism for—or capacity to activate—the RXFP-1 receptor. Interestingly, one of the most remarkable differences between serelaxin and pRLX is the extension of the C- and N-terminal portions of both chains A and B that protrude from the central polycyclic region determined by the three disulfide bridges. We speculate that these terminal regions might be more exposed to enzymatic cleavage, as compared to the central portion of the molecule, since it is known that, generally speaking, the presence of disulfide bonds confers to the peptide not only high rigidity and low flexibility but also resistance against enzymatic cleavage of susceptible peptide bonds [24]. Accordingly, the different lengths and compositions of the terminal sequences of the two relaxin molecules may result in different susceptibilities to metabolism. Taken together, the present findings point to pRLX as a feasible candidate drug for such oral formulations, owing to its higher resistance to gut peptidases than serelaxin, which can override its lower affinity to RXFP-1 with respect to serelaxin [18]. This intriguing possibility is also supported by additional data from the literature, indicating that: (i) pRLX has been shown to possess similar in vivo bioactivity as H2 RLX by the classical mouse pubic symphysis relaxation assay [25] and in vitro platelet inhibition assay [17], (ii) another peptide drug, the glucagon-like peptide (GLP)-1 analog exanetide used in type-2 diabetes, has been developed by a heterologous GLP-1-like molecule found in reptilian salivary secretion, which shows a 53% amino acid sequence homology with the mammalian peptide, allowing it to exert glucoregulatory actions mediated by the GLP-1 receptor but, at variance with true GLP-1, is resistant to peptidase degradation due to a key amino acid difference in position 3 and has a substantially longer half-life [26]. However, further studies by means of ex vivo or in vivo gut absorption models [21] are undoubtedly needed to substantiate the hypothesis of using pRLX-based formulations by the oral route. Moreover, while serelaxin has been extensively studied in clinical trials up to phase III, pRLX was only employed in pilot clinical studies, and its use as a human drug has been discontinued since 1980s [27], thus requiring much work and costs to be developed for such purposes.

It must be underlined that digestion drawbacks are common in most therapeutic peptides and should not be regarded as an absolute contraindication. In the search for possible solutions, various strategies have been investigated. A meaningful approach is to enhance permeation through the intestinal barrier in order to allow sufficient amounts of proteins to be absorbed before being digested [21]. While co-administration of permeation enhancers to weaken the intestinal barrier is no more in use because of toxicity concerns [28], good promise is offered by mucoadhesive nanoformulations capable of both protecting the proteins from digestive enzymes and prolonging the contact with the intestinal surface epithelium and the lymphatic Peyer’s patches scattered along the gut mucosa [29]. In particular, Peyer’s patches are considered the major route for protein absorption since their epithelial lining is specialized to allow passage of intact macromolecules by a trans-epithelial vesicular mechanism [30]. On the other hand, should efficacious RLX nanoformulations suitable for oral delivery be eventually identified and produced, another level of issues should be considered related to individual pathophysiological variables which can influence absorption, namely gastrointestinal motility and secretion, concurrent gastrointestinal and liver disorders, etc. Taken together, all the above factors may likely render the pharmacokinetic properties of orally given RLX, largely different from those known upon parenteral delivery [7]. A positive note is that at variance with oral insulin, which must be absorbed in reproducible quantities at defined time points after administration to achieve effective post-prandial metabolic control, RLX should not meet these stringent requirements to exert its biological/therapeutic effects.

## 4. Materials and Methods

### 4.1. Materials

Serelaxin (batch B917056/1/1, prepared by Boehringer-Ingelheim Inc., Wien, Austria) was kindly donated by the Relaxin RRCA Foundation, Florence, Italy. Parallel experiments were carried out using highly purified luteal porcine RLX (pRLX, ~3000 U/mg), kindly donated by Dr. Antonio Luigi Scarpa, formerly at IBSA Institut Biochimique, Lugano, Switzerland. Stock aliquots of both peptides (500 μg/mL) were stored at −80 °C and thawed immediately before use. Throughout the experiments, silicon-coated test tubes were used to prevent the adhesion of the peptides to the walls.

Unless otherwise stated, all chemicals and reagents used in the experiments were from Merck-Sigma-Aldrich (Milan, Italy), while cell culture plasticware was from VWR-Avantor (Milan, Italy).

### 4.2. Simulated Intestinal Digestion

The effects of duodenal enzyme digestion on the noted serelaxin and pRLX preparations was carried out by addition of a simulated intestinal fluid (SIF) [24], composed of a solution of pancreatin (500 μg/mL), CaCl_2_ 0.75 M (0.1 *v*/*v*), pH adjusted to 7 adding dropwise an aqueous solution of NaOH 0.1 M to pRLX and serelaxin solutions. In particular: 0.5 mg/mL aqueous solution of pRLX (MW 5918, 0.084 mM) was added to SIF in a 50 mL Falcon, mixing in a 37 °C bath, to a final pRLX concentration 1.12 μM; 0.2 mL of this solution were sampled and quenched by pH adjusting (0.1 M HCl), at the following times: t = 0”/10”/20”/30”/60”/120”/600”/1800”. 5 mg/mL aqueous solution of serelaxin (MW 5963, 0.84 mM) was added to SIF in a 50 mL Falcon, mixing in a 37 °C bath, to a final pRLX concentration 1.12 μM; 0.2 mL of this solution were sampled and quenched by pH adjusting (0.1 M HCl), at the following times: t = 0”/10”/20”/30”/60”/120”/600”/1800”. Both supernatant sampled solutions were collected after centrifugation at 10,000 rpm for 10′. Blank sampled solutions were prepared by mixing SIF in a 50 mL Falcon in a 37 °C bath and sampling at the above-described times. Finally, the samples were quickly frozen and stored at −80 °C until further use.

### 4.3. HPLC Analysis

Chromatographic investigation was performed using an UHPLC Ultimate 3000 (Thermo Scientific, Waltham, MA, USA), equipped with a variable wavelength detector, in the following conditions: Stationary phase: C18 Acquity BEH column (130 Å, 1.7 μm, 2.1 × 150 mm; Waters, Milford, MA, USA), temperature: 60 °C, flow: 0.5 mL/min; eluents: 0.1% (*v*/*v*) TFA in H_2_O (A) and 0.1% (*v*/*v*) TFA in CH_3_CN (B); gradient: 10–90% B in A in 15 min; λ: 215 nm.

### 4.4. Cell Culture and cAMP Assay

Human monocytic THP-1 cells (European Collection of Cell Cultures ECACC, Salisbury, UK) were cultured in suspension in RPMI medium containing 10% fetal calf serum, 250 U/mL penicillin G and 250 μg/mL streptomycin in a 5% CO_2_ atmosphere at 37 °C. Aliquots containing 6 × 10^5^ THP-1 cells in 1 mL PBS were placed in a 24-well plate, added with IBMX (100 μM) to prevent cAMP catabolism, and then incubated with intact serelaxin (100 ng/mL), pRLX (100 ng/mL) or equivalent solutions of their digestion by-products. According to previous studies [16,17], cAMP assay was carried out 15 min after addition of the stimuli, i.e., in the midst of the second, sustained cAMP surge upon RXFP1 stimulation. Measurements of cAMP were determined by the chemiluminescent HitHunterXP™ cAMP II assay (DiscoverX, Birmingham, UK). The standard curve was performed in duplicate, and the experimental points in triplicate. Since the standard curve was sigmoidal, nonlinear regression was used to extrapolate unknown experimental cAMP values, expressed as nmol/L, according to the manufacturer’s instructions. Two independent experiments were performed. For positive control, the adenylate cyclase activator forskolin (10^−4^ M) was used. Finally, the values of cAMP in each experimental sample were normalized by the amount of proteins, measured by the micro-BCATM Protein Assay Kit (Pierce, IL, USA) method, and expressed as mg/mL.

## Figures and Tables

**Figure 1 ijms-24-00048-f001:**
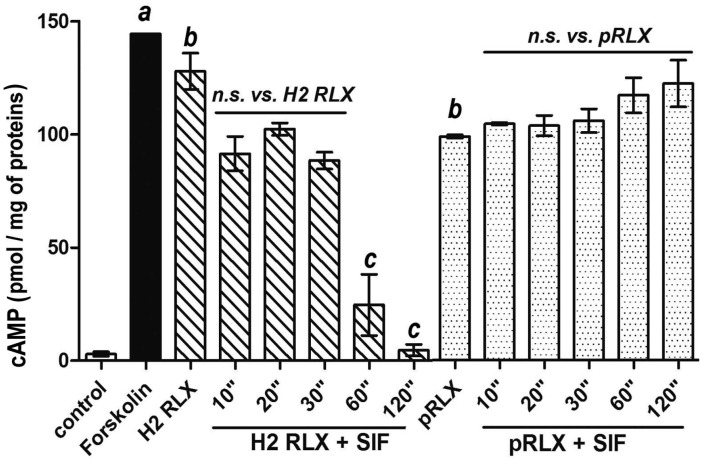
cAMP response by THP-1 cells upon stimulation with intact serelaxin (H2 RLX) and porcine relaxin (pRLX) (100 ng/mL) or their digestion products obtained by incubation with SIF for the reported times. Forskolin (10^−4^ M) was used as positive control to evaluate the maximal cAMP levels. The H2 RLX digestion products show a progressive decrease in cAMP generation, which correlate with the SIF exposure time, whereas the pRLX digestion products stimulate the cAMP generation at similar levels as the intact peptide at all times tested (up to 2 min). Significance of differences: a,b *p* < 0.001 vs. control; c, *p* < 0.001 vs. H2 RLX; n.s., not significant.

**Figure 2 ijms-24-00048-f002:**
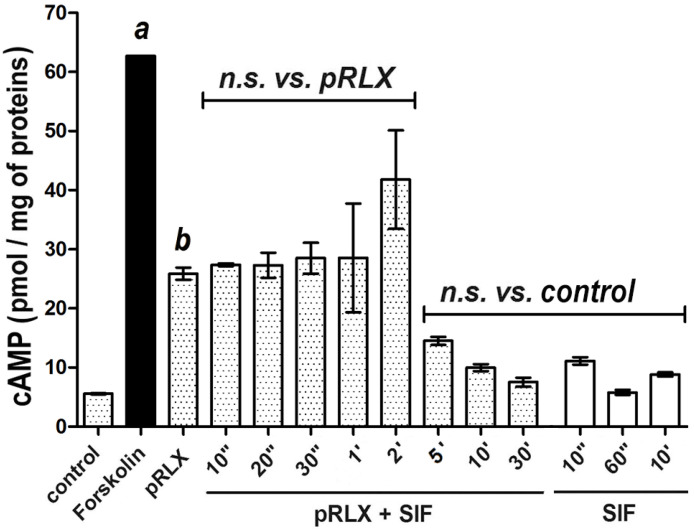
cAMP response by THP-1 cells upon stimulation with porcine relaxin (pRLX) (100 ng/mL) or its digestion products obtained by SIF incubation for up to 30 min. Forskolin (10^−4^ M) was used as positive control to evaluate the maximal cAMP levels. A sudden decrease in cAMP generation can be observed at a time point comprised between 2- and 5-min. SIF alone was devoid of any cAMP-stimulating effects on THP-1 cells. Significance of differences: a,b *p* < 0.001 vs. control.

**Figure 3 ijms-24-00048-f003:**
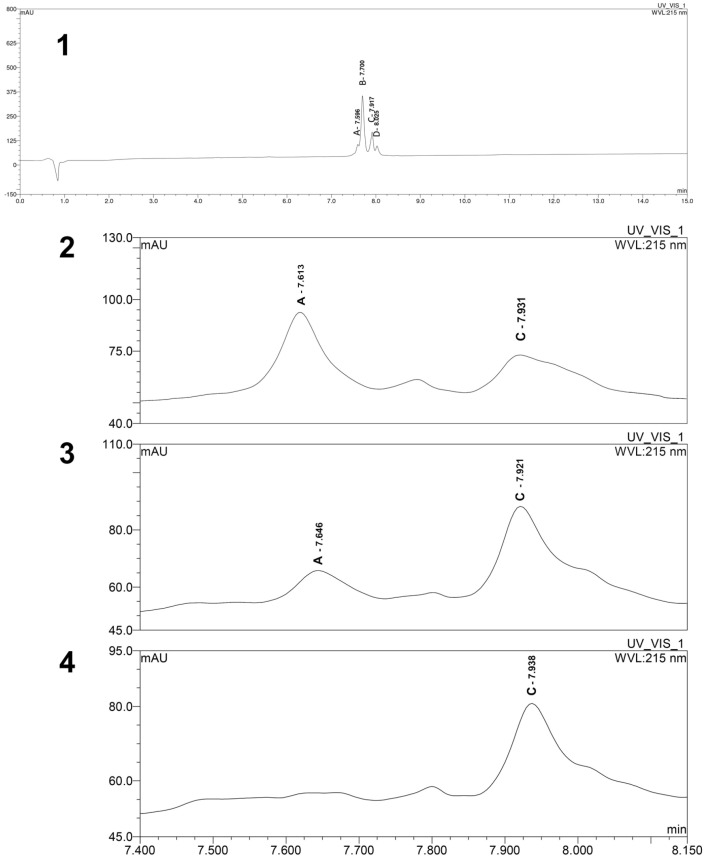
Time course of pRLX simulated digestion. HPLC profile of pRLX at t = 0 min (chromatographic conditions reported in the text), showing the four peaks labeled A, B, C, and D (panel (**1**)). Profile of the relevant peaks after SIF treatment, t = 10” (panel (**2**)), t = 2 min (panel (**3**)), and t = 5 min (panel (**4**)).

## Data Availability

Not applicable.

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
