# Peer review of "Porcine Relaxin but Not Serelaxin Shows Residual Bioactivity after In Vitro Simulated Intestinal Digestion—Clues for the Development of New Relaxin Peptide Agonists Suitable for Oral Delivery"

_ijms, 2022, doi:10.3390/ijms24010048_

Round 1

Reviewer 1 Report

Review report

The article titled: “Porcine relaxin but not serelaxin shows residual bioactivity after in vitro simulated intestinal digestion. Clues for the development of new relaxin peptide agonists suitable for oral delivery” was aimed to investigate the feasibility of oral administration of serelaxin and purified porcine RLX  using an in vitro model to simulate the possible behaviour of gastroprotected formulations subjected to gut enzymatic digestion.

 Materials and methods part

4.1. Materials

·       CH should be replaced with Switzerland

Results

·       Used cells in Materials and methods part were wrote out as THP-1 cells, but in Result part we see the cells wrote as THP-1 cells – A. Please, specify

In summary, I express a high degree of doubt that the indicated increased time of activity of pRLX is sufficient for its therapeutic action to take place. The authors should show examples and discuss the measurability of this resistance with the possibility of the substance to effectively affect the cardiovascular system. Otherwise, although accurately conducted, the present study would not be very useful.

My opinion is that the article can be reconsidered after major revision

Author Response

This reviewer highlighted a key point which came out from our study, namely that the reported in vitro finding cannot predict whether pRLX could be effective in vivo upon oral administration. However, the reported differences in resistance to gut enzyme digestion between the human and porcine peptides open a path to further research focused on the residual pRLX fragment which, owing to its improved resistance to digestion, could be exploited for the design and synthesis of new agonists suited for oral administration. In a recent paper, we provided evidence that H1 RLX analogues can be stabilized to induce the appropriate 3-D conformation by intra-chain tri-azolic staples, which should further enhance their resistance to digestive enzymes (D'Ercole et al.. Front Pharmacol. 2022 Aug 11;13:942178): this is an intriguing subject that we aim at exploring soon. This information has been added to the Discussion (page 5, lines 174-178)

The suggested corrections in the Methods and Results have been made: thank you for accurate reading.

Reviewer 2 Report

In this paper the authors present experimental work about the bioactivity of porcine relaxin and serelaxin, after in vitro simulated intestinal digestion. The obtained results show that, for the former it is possible to retain significant bioactivity up to two minutes. 

With some minor exceptions, the manuscript is clearly written and carefully referenced. The authors have described well what has been reported so far in the literature and make proper comparisons to the work presented.

However, there is one point that should be made clear.  If the authors want to stress the possible suitability for oral delivery one would expect that the simulated digestion would cover digestion by salivaand simulated gastric digestion in addition to the SIF treatment done.

Finally, figure 3 is of very low quality. It would be very helpful if panels 2, 3 and 4 could be plotted vertically sharing the same x-axis, and having equivalent y-axis, to fully compare the chromatographic peaks. This could be a new figure. In panel 1, I would expect to have the injection marked, unless this corresponds to 0.0 min (which it does seems to be the case).

Author Response

We are grateful to this reviewer for kind words of appreciation of our work and for valuable suggestions to improve it. The following changes have been made to the revised version of our manuscript.

  1. Figure 3 has been replaced with a new one at higher resolution, as suggested. As noted by the reviewer, the injection time is at time zero: thus, we chose not to insert any mark on the chromatogram for easy reading.
  2. We agree with the reviewer that this is a key issue, since identification of the amino-acidic sequence of peak A can be the logical conclusion of our study. We did try and perform a mass spectrometry analysis of our pRLX batch used for the digestion experiments, but we got inconclusive results which did not allow us to identify or predict the actual sequence of peak A with a reasonable degree of precision. For this reasons, we prefer not to include these data in the present manuscript but to use them as a starting point for a future study dedicated to identify this peptide and modify it to try and obtain a functional RXFP1 agonist suitable for oral administration.
  3. We have extended the discussion as suggested trying to highlight which molecular differences between the two relaxins could help explain their different resistance to SIF digestion. However, as we stated in the previous point, we have no data on the possible sequence of peak A and we lack solid arguments to provide a convincing and complete explanation of this finding. It is worth noting that a similar behaviour has been reported for other heterologous bioactive peptides, for instance calcitonin and GLP-1, possibly due to evolutionary divergences between peptidic substrates and proteolytic enzymes in different species. This point has been addressed to in the Discussion (pages 5-6, lines 183-195).
  4. Throughout the text, the needed corrections to the formulas have been made: thank you for careful reading.

Reviewer 3 Report

The article: “Porcine relaxin but not serelaxin shows residual bioactivity after in vitro simulated intestinal digestion. Clues for the development of new relaxin peptide agonists suitable for oral delivery” written by Lorenzo Pacini and co-authors presents interesting approach to evaluate in vitro the enzymatic degradation bioactive peptides hormone serelaxin and porcine relaxin. The numerous previous studies have shown cardioprotective effects of relaxin, which become potential therapeutic in cardiovascular disease. The many  clinical trials for the use of relaxin in acute heart failure were conducted. One of inconvenience is route of administration to patient. Generally peptides are sensitive to enzymatic digestion and it is one of major reason to oral unavailability as drug. Therefore, the research for new ways of administering peptide-drugs, taking into account the preservation of their biological function, is an extremely important and modern issue. These studies can directly improve the quality of life of patients who need such treatment. Moreover this methodology could be used to examine oral administration route for other peptide-drugs. The manuscript has been prepared with due attention to detail. Both the introduction and discussion have been written in concise scientific language. Only few minor editorial bugs need correction. The research results and conclusions are drew accurate and well presented. However, the article needs a few minor corrections, which are and do not diminish the research value, listed below:

1.      The Resolution of Figure 3 have to be higher, the quality of chromatography plot would be more readable. Moreover, the panel 1 could be focus only at peaks area.  

2.      Could you please explain: Why did not You make a identify analysis of contains of each fraction and especial for peak A? The mass spectrometry it would be fast, the most useful and suitable method, which I recommend. The knowledge of direct sequence responsible for interaction with receptors is crucial in further considerations.

3.      The discussion needs to be extended to explain what structural aspects of pRLX contribute to its better proteolytic resistance. What are molecular differences between pRLX and serelaxin? They could be only teoretical considerations, based on available scientific literature.

4.      The stoichiometric coefficients in the formulas of inorganic compounds should be written with appropriate subscripts index, e.g. (line 241, 254).

Author Response

We are grateful to this reviewer for kind words of appreciation of our work and constructive criticisms. The following changes have been made to the revised version of our manuscript.

1. Indeed, the issue of salivary and gastric fluid digestion of peptide drugs can be easily prevented by proper pharmaceutical formulations withstanding the oral and gastric environment. This caution does not affect pharmacokinetics, because the main site for drug absorption is the small bowel, especially lymphatic Peyer’s patches. This is the reason why we focused on gut digestion, as this is the place where a putative oral formulation has to dissolve its gastroresistant envelope and release the relaxin. This point has been reported in the Discussion (page 5, lines 155-158).

2. We appreciate the suggestion about the arrangement of panels in Figure 3: this has been rebuilt by stacking panels 2-4 with common x-axis, as suggested, and submitted at a higher resolution.

Round 2

Reviewer 1 Report

All the questions I asked have been answered